# Targeted Alpha Therapy: Progress in Radionuclide Production, Radiochemistry, and Applications

**DOI:** 10.3390/pharmaceutics13010049

**Published:** 2020-12-31

**Authors:** Bryce J. B. Nelson, Jan D. Andersson, Frank Wuest

**Affiliations:** 1Department of Oncology, University of Alberta, 11560 University Ave, Edmonton, AB T6G 1Z2, Canada; bjnelson@ualberta.ca (B.J.B.N.); janderss@ualberta.ca (J.D.A.); 2Edmonton Radiopharmaceutical Center, Alberta Health Services, 11560 University Ave, Edmonton, AB T6G 1Z2, Canada

**Keywords:** targeted alpha therapy, alpha particle therapy, targeted radionuclide therapy, theranostics, actinium-225, bismuth-213, astatine-211, radium-223, thorium-227, terbium-149

## Abstract

This review outlines the accomplishments and potential developments of targeted alpha (α) particle therapy (TAT). It discusses the therapeutic advantages of the short and highly ionizing path of α-particle emissions; the ability of TAT to complement and provide superior efficacy over existing forms of radiotherapy; the physical decay properties and radiochemistry of common α-emitters, including ^225^Ac, ^213^Bi, ^224^Ra, ^212^Pb, ^227^Th, ^223^Ra, ^211^At, and ^149^Tb; the production techniques and proper handling of α-emitters in a radiopharmacy; recent preclinical developments; ongoing and completed clinical trials; and an outlook on the future of TAT.

## 1. Introduction

Radionuclide therapy has been employed frequently in the past several decades for disease control, curative therapy, and pain management applications [1]. Targeted radionuclide therapy (TRT) is advantageous as it delivers a highly concentrated dose to a tumor site—either directly to the tumor cells or to its microenvironment—while sparing the healthy surrounding tissues. It has been clinically demonstrated using a variety of radionuclides to treat malignancies, including polycythemia, cystic craniopharyngioma, hyperthyroidism, synovitis and arthritis, and numerous cancers, such as thyroid cancer, bone tumors and metastasis, hepatic metastasis, ovarian cancer, neuroendocrine tumors, leukemia, lymphoma, and metastatic prostate cancer [2,3,4]. Since radionuclide therapy targets diseases at the cellular level, it has advantages for treating systemic malignancies such as tumor metastases over other forms of therapy such as external beam therapy, where full body irradiation is impossible. In addition to being minimally invasive, radionuclide therapy can be shorter in duration than chemotherapy [2].

In TRT, therapeutic radionuclides including alpha (α), beta (β^−^), and Auger electron emitters are typically conjugated to a targeting vector such as monoclonal antibodies, biomolecules, peptides, nanocarriers, and small-molecule inhibitors. To maximize the therapeutic efficacy of a TRT radiopharmaceutical, its radionuclide decay pathway, particle emission range, and relative biological effectiveness should be matched appropriately for a given tumor mass, size, radiosensitivity, and heterogeneity [1].

This review focuses on targeted alpha therapy (TAT), outlined by the graphical abstract in Figure 1. A detailed overview of α-emitting radionuclides currently employed in radiotherapy is presented and compared to radionuclides with different emissions, including β^−^ particle and auger electron emitters. Production techniques for α-emitters are outlined, including their separation and unique handling requirements, followed by their radiochemistry and targeting characteristics. Preclinical developments and clinical applications of α-emitters are discussed along with current limitations, potential areas for improvement, and anticipated applications.

## 2. Selecting Radionuclides for Radiotherapy

When selecting a radionuclide for clinical application, the physical and biochemical characteristics must be considered. Physical characteristics include physical half-life, type of emissions, energy of the emissions, daughter products, method of production, and radionuclidic purity. Biochemical characteristics include tissue targeting, retention of radioactivity in the tumor, in vivo stability, and toxicity [2]. For radiotherapy, it is desirable to have a high linear energy transfer (LET), where there is a high ionization energy deposited per unit length of travel. Radionuclides with a high LET deposit radioactive emission energy within a small range of tissue, thereby sparing surrounding healthy tissue and keeping the radioactive dose within, as much as possible, the patient’s organ to be treated. It can also be advantageous for the therapeutic radionuclide, or a complementary theranostic radionuclide, to emit positrons (β^+^) or gamma (γ) radiation. This enables positron emission tomography (PET) or single photon emission tomography (SPECT) imaging and visualization of radiopharmaceutical distribution within a patient’s body, permitting treatment monitoring. Table 1 outlines key characteristics of α, β^−^, and auger electron emitters, and some clinical applications for cancer TRT that have been explored.

β^−^ emitting radioisotopes have a relatively long pathlength (≤12 mm) and a lower LET of ~0.2 keV/μm, giving them effectiveness in medium–large tumors [1]. However, they lack success in solid cancers with microscopic tumor burden. This may be attributed to their emissions releasing the majority of their energy along a several millimeter long electron track, irradiating the surrounding healthy tissue instead of depositing their main energy into the micro-metastatic tumor cells where the radionuclide was delivered [6].

Clinical success has been demonstrated with the β^−^ emitters ^90^Y and ^131^I conjugated with anti-CD20 monoclonal antibodies in follicular B-cell non-Hodgkin lymphoma [6], ^177^Lu-labeled prostate-specific membrane antigen (PSMA) peptides in metastatic, castration-resistant prostate cancer (CRPC) and ^177^Lu-DOTATATE for neuroendocrine tumors [7,8].

Auger electrons have a medium LET (4–26 keV/μm) [1]; however, their short pathlength of 2–500 nm limits the majority of their effects to within single cells, requiring the radionuclide to be transported into the cell and preferably incorporated into DNA to achieve high lethality. They can also kill cancer cells by damaging the cell membrane and kill non-directly targeted cells through a cross-dose or bystander effect [9]. Clinical studies with Auger electrons for cancer therapy have been limited; however, some encouraging results were obtained using [^111^In]In-DTPA-octreotide in rats with pancreatic tumors, [^125^I]I-IUdR where tumor remissions were achieved, and [^125^I]I-mAb 425 where the survival of glioblastoma patients improved [5,10,11]. However, it has also been determined that some Auger electron emitting compounds, such as [^123^I]I-IUdR and [^125^I]I-IUdR only kill cells in the S-phase of the cell cycle, highlighting a potential treatment limitation [12].

α-particles have a high LET (80 keV/μm) and a moderate pathlength (50–100 μm), giving them an effective range of less than 10 cell diameters. This makes them suitable for microscopic tumor cell clusters, while sparing normal organs and surrounding healthy tissues. Importantly, α-particle lethality is not dependent on the cell cycle or oxygenation, and the DNA damage is often via double strand and DNA cluster breaks and is therefore much more difficult to repair than β^−^ damage [6]. It has been estimated that to attain a single cell kill probability of 99.99%, tens of thousands of β^−^ decays are required, whereas only a few α-decays at the cell membrane achieves the same kill probability [13]. From this, it has been estimated that one α particle transversal can kill a cell [14]. Most α-emitters are conjugated to a wide range of targeting vectors for delivery to their target site, though some have intrinsic targeting properties, such as the affinity of ^223^Ra-dichloride for bone [1]. Preclinical and clinical studies using α-emitters have been ongoing for a variety of cancers, some of which include recurrent brain tumors, recurrent ovarian cancers, human epidermal growth factor receptor-2 (HER-2) positive cancers, myelogenous leukemia, non-Hodgkin lymphoma, metastatic melanoma, and skeletal metastases in prostate cancer [6]. Of these, the most theranostic research is performed on prostate and neuroendocrine tumors (NETs). Examples of studies are numerous—one preclinical study using [^212^Pb]Pb-trastuzumab found a single injection reduced tumor growth by 60–80%, reduced aortic lymph node metastasis, and prolonged survival or tumor-bearing mice [15]. Another study outlined how α-particle radiotherapy for metastatic castration resistant prostate cancer using [^225^Ac]Ac-PSMA-617 was able to overcome resistance to [^177^Lu]Lu-PSMA-617 β^−^-particle therapy [16]. Additionally, a study using [^213^Bi]Bi-DTPA and [^213^Bi]Bi-DOTATATE in mice resulted in a factor of six increase in cell killing compared to [^177^Lu]Lu-DOTATATE [17,18]. These studies highlight the clinical importance and potential of α-emitters, and their potential to be more efficient and effective than β^−^- therapy.

## 3. Alpha Emitter Decay Properties

As emissions from radioactive decay, α-particles are naked ^4^He nuclei with a +2 charge. They are 7300 times larger than the mass of β^−^ and Auger electrons, giving them significant emission momentum that reduces deflection and results in a near-linear emission path, as opposed to the winding path of β^−^ particles. With an emission kinetic energy between 5–9 MeV, coupled with a particle range of 50–100 μm, this classifies α-particles as high LET. The energy distribution between the alpha particle and the recoiling daughter atom is typically 98% to 2%. Upon decay, energy imparted to the daughter recoil atom can reach 100 keV [19], which is far higher than the binding energy of the strongest chemical bonds, resulting in release of the daughter isotope from its targeting vector. An example is ^219^Rn, which has a daughter recoil range of 88 nm in a cellular environment [19]. These daughters often have a serial decay chain with their own α-emitting progeny, leading to untargeted irradiation of surrounding tissues. As a result, only a limited number of α-emitting radioisotopes are suitable for therapy due to their decay characteristics. The half-life of the radionuclide should be reasonable for therapy; it should not be too short to allow sufficient time for production, radiopharmaceutical synthesis, and delivery to the patient, and it, as well as the half-life of any daughter radionuclide, should not be too long to avoid excess patient dose.

The recoil energy caused by the decay of α-emitters invariably destroys α-emitter-targeting vector chemical bonds, often releasing α-emitting progeny with different chemistries that can lead to undesirable toxicities. The presence of γ-ray emissions in an α-emitter decay chain is also of interest for imaging purposes. Therefore, it is important to understand the half-lives, emissions, and decay characteristics when selecting clinically relevant α-emitters. Figure 2 depicts decay chains that contain some common therapeutic α-emitters, and Table 2 outlines the decay characteristics of some notable α-emitters used in α therapy, including their daughters, half-lives, decay energies, and emissions.

^225^Ac (t_1/2_ = 9.9 d, 5.8 MeV α particle) decays to ^209^Bi with six intermediate radionuclide progenies. These daughters include ^221^Fr (t_1/2_ = 4.8 min; 6.3 MeV α particle and 218 keV γ emission), ^217^At (t_1/2_ = 32.3 ms; 7.1 MeV α particle), ^213^Bi (t_1/2_ = 45.6 min; 5.9 MeV α particle, 492 keV β^−^ particle and 440 keV γ emission), ^213^Po (t_1/2_ = 3.72 μs; 8.4 MeV α particle), ^209^Tl (t_1/2_ = 2.2 min; 178 keV β^−^ particle), ^209^Pb (t_1/2_ = 3.23 h; 198 keV β^−^ particle) and ^209^Bi (stable). From this, a single ^225^Ac decay yields a total of four α, three β^−^ disintegrations, and two γ emissions, which classifies ^225^Ac as a “nanogenerator” or “in vivo generator”. Therefore, the 9.9 d half-life of ^225^Ac, the multiple α particle emissions in its decay chain, and its rapid decay to ^209^Bi make ^225^Ac an attractive candidate for TAT [21]. The γ emissions would be useful for SPECT imaging of in vivo radiopharmaceutical distribution, giving the ^225^Ac decay series theranostic potential; however, due to the potency of ^225^Ac, the small administered doses and correspondingly low γ emissions would make planar SPECT imaging difficult [21]. Of note, the intermediate ^213^Bi possesses attractive potential and can be separated from the ^225^Ac decay series for use. However, the short 45.6 min half-life of ^213^Bi presents challenges for processing, radiolabeling, and radiopharmaceutical administration, resulting in a limited time in circulation to accumulate at its target site and achieve its intended therapeutic effects.

^224^Ra (t_1/2_ = 3.63 d, 5.7 MeV α particle, 241 keV γ emission) decays to ^208^Pb with six intermediate radionuclide progenies. These daughters include ^220^Rn (t_1/2_ = 55.6 s, 6.3 MeV α particle), ^216^Po (t_1/2_ = 0.15 s, 6.8 MeV α particle), ^212^Pb (t_1/2_ = 10.6 h, 93.5 keV β^−^ particle, 238 keV γ emission), ^212^Bi (t_1/2_ = 60.6 min, 6.1 MeV α particle, 834 keV β^−^ particle, 727 keV γ emission), ^212^Po (t_1/2_ = 0.30 μs, 8.8 MeV α particle), ^208^Tl (t_1/2_ = 3.1 min, 650 keV β^−^ particle, 2614 keV γ emission), and ^208^Pb (stable). From this, a single ^224^Ra decay yields a total of four α particles, two β^−^ disintegrations, and six γ emissions, also classifying ^224^Ra as a “nanogenerator”. The bone-seeking properties of ^224^Ra and its favorable half-life has resulted in its use in α-therapy, and its intermediates ^212^Pb and ^212^Bi show potential for TAT, with ^212^Pb preferable to ^212^Bi for administration due to the longer half-life of ^212^Pb, permitting more dose from its ^212^Bi progeny to be delivered [1].

^227^Th (t_1/2_ = 18.7 d, 6.0 MeV α particle, 236 keV γ emission) decays to ^207^Pb with six intermediate radionuclide progenies. These daughters include ^223^Ra (t_1/2_ = 11.4 d, 5.7 MeV α particle, and 269 keV γ emission), ^219^Rn (t_1/2_ = 3.96 s, 6.8 MeV α particle, 271 keV γ emission), ^215^Po (t_1/2_ = 1.78 ms, 7.4 MeV α particle), ^211^Pb (t_1/2_ = 36.1 min, 471 keV β^−^ particle, 404 keV γ emission), ^211^Bi (t_1/2_ = 2.14 min, 6.6 MeV α particle, 172 keV β^−^ particle, 351 keV γ emission), ^207^Tl (t_1/2_ = 4.77 min, 492 keV β^−^ particle), and ^207^Pb (stable). ^227^Th and ^223^Ra are both nanogenerators, releasing up to four α particles during the decay chain, and their γ emissions allow for imaging [1].

^211^At (t_1/2_ = 7.2 h, 5.9 MeV α particle) decays to ^207^Pb with two intermediate radionuclide progenies in separate paths. These daughters include ^207^Bi (t_1/2_ = 31.6 y, electron capture) which decays to ^207^Pb and ^211^Po (t_1/2_ = 0.52 s, 7.5 MeV α particle, Kα x-rays) which decays to ^207^Pb. The decay to ^211^Po would permit in vivo imaging of ^211^At using the emitted Kα x-rays.

^149^Tb (t_1/2_ = 4.1 h, 4.0 MeV α particle, 638 keV β^+^ particle), decays to ^149^Sm and ^145^Nd in two separate paths. In one path, its daughters include ^149^G (t_1/2_ = 9.28 d), ^149^Eu (t_1/2_ = 93.1 d, electron capture), and ^149^Sm (stable). The other path includes ^145^Eu (t_1/2_ = 5.9 d, 740 keV β^+^ particle, 894 keV γ emission), ^145^Sm (t_1/2_ = 340.3 d, electron capture, 61 keV γ emission), ^145^Pm (t_1/2_ = 17.7 y, electron capture, 72 keV γ emission), and ^145^Nd (stable) [22]. The decay scheme for ^149^Tb is quite favorable since it releases short-range α particles from only one radionuclide, with complementary γ emissions and positrons that can be employed for imaging purposes in an “alpha-PET” combination [23,24]. Having only one α-emitter in its decay scheme implies a minimal toxicity from daughter recoil during radioactive decay, which should reduce excessive dose burden [25].

## 4. Alpha Emitter Production, Separation, and Handling

Common production methods for clinically relevant α-emitters can involve a cyclotron that accelerates and bombards a target using variety of particles, including protons, alpha particles, lithium and carbon ions, and nuclear reactors such as fast breeder reactors [26]. α-emitters are often delivered by a generator, where a parent isotope decays to the desired radionuclide that is then extracted. Current, anticipated, and potential production methods for various α-emitters are outlined in Table 3.

### 4.1. ^225^Ac and ^213^Bi

^225^Ac and ^213^Bi are currently produced from ^229^Th generators (t_1/2_ = 7397 y). ^229^Th is sourced from the decay of fissile ^233^U, which was originally produced by neutron irradiation of natural ^232^Th. These generators can be milked over a 3-week period, permitting the separation of ^225^Ra and ^225^Ac. Current sources of ^229^Th that allow production of clinically sufficient activities of ^225^Ac/^225^Bi are available at the Directorate for Nuclear Safety and Security of the Joint Research Centre (JRC) of the European Commission in Karlsruhe, Germany, Oak Ridge National Laboratory (ORNL), the Institute of Physics and Power Engineering (IPPE), and recently Canadian Nuclear Laboratories. Since the 1990s, JRC has produced approximately 13 GBq annually, while ORNL has produced approximately 22 GBq/year, with these ^225^Ac products found safe for human administration and applied for patient treatment [32]. As of 2018, worldwide production of ^225^Ac has been 68 GBq/year, and while this does support preclinical studies, it only supports several hundred patients per year (when labeled with a 4–50 MBq dose) and prevents large scale ^225^Ac/^213^Bi generator production. However, no production technique has emerged as an effective and economic solution, leaving ^225^Ac supply as a patchwork of production methods with varying yields and radioisotopic purities [33]. It is clear that alternative methods of ^225^Ac production would enhance the supply chain and enable more widespread use.

Alternative production methods of ^225^Ac include neutron, proton, and deuteron irradiation of ^226^Ra targets, and high-energy proton irradiation of ^232^Th targets. Large-scale production of ^225^Ac by cyclotron proton irradiation of ^226^Ra via the ^226^Ra(p,2n)^225^Ac reaction shows promise. There is a significant 710 mb cross-section peak at 16.8 MeV proton energy, making this ideal for high-yield and cost-effective production for cyclotrons with beam energies less than 20 MeV [26]. Additionally, no long-lived actinium byproducts (^227^Ac, t_1/2_ = 21.8 y) are produced, and coproduced ^226^Ac (t_1/2_ = 29 h) via the ^226^Ra(p,n)^226^Ac and ^224^Ac (t_1/2_ = 2.8 h) via the ^226^Ra(p,3n)^224^Ac can be minimized by optimizing proton beam energy, and allowing for a “cool-down” period for decay post-irradiation due to their shorter half-lives compared to ^225^Ac. Although handling ^226^Ra is technically demanding due to its inherent radioactivity and 1600-year half-life, this method is preferred for large-scale, cost-effective production [32]. Irradiating ^232^Th with high-energy protons has been demonstrated to produce several GBq of ^225^Ac using an intense proton beam irradiation for 10 days. However, this method produces a variety of radionuclidic impurities that must be removed by chemical separation and significant amounts of long-lived ^227^Ac (0.1–0.2% at end of bombardment) [32].

^213^Bi generators have been developed by JRC and demonstrated to reliably prepare up to 2.3 GBq of ^213^Bi radiolabeled compounds for therapeutic doses when loaded with up to 4 GBq of ^225^Ac [32]. They feature homogenous distribution of the ^225^Ac over the generator resin, minimizing radiolytic degradation and permitting operation for several weeks.

However, the availability of ^213^Bi generators depends on the availability ^225^Ac generators, which are already in short supply [17].

### 4.2. ^224^Ra, ^212^Bi, and ^212^Pb

^224^Ra, ^212^Pb, and ^212^Bi can be produced from ^228^Th (t_1/2_ = 1.92 years) generators; however, radiolytic damage occurred to the Na_2_TiO_3_ resin, resulting in diminished yield and posing a radiation safety issue [27]. This resulted in a switch to ^224^Ra (t_1/2_ = 3.6 d)-based generators from which ^212^Bi and ^212^Pb are obtained by elution. Despite producing high yields of ^212^Pb (>90% of expected activity per daily elution) and its daughter ^212^Bi, the generator must be replaced after 1–2 weeks due to the short half-life of ^224^Ra [34].

### 4.3. ^227^Th and ^223^Ra

^227^Th and ^223^Ra are available by separation from their parent ^227^Ac radionuclide. Production of ^223^Ra for clinical applications utilizes ^227^Ac/^227^Th generators, where the parent isotopes are on actinide chromatographic resins, and ^223^Ra-chloride solution is obtained after elution and purification on a cation exchange column [35].

### 4.4. ^211^At

^211^At can be produced using an α-particle beam to bombard natural bismuth at 28–29.5 MeV via the ^209^Bi(α,2n)^211^At reaction [29]. Despite being a straightforward method of production, the number of accelerators capable of a 28 MeV α beam limits the availability of ^211^At. ^211^At can also be produced via the ^209^Bi(^7^Li,5n)^211^Rn reaction, where ^211^Rn (t_1/2_ = 14 h) decays to ^211^At. This method, currently under development, exploits the longer half-life of ^211^Rn to extend the timeframe for effective distribution and use of ^211^At [30].

### 4.5. ^149^Tb

The availability of ^149^Tb is currently quite limited primarily due to inaccessible production routes. ^149^Tb has been produced for use in preclinical α-therapy studies by the isotope and separation online facility (ISOLDE) at CERN, using high-energy proton irradiation of tantalum targets (1.4 GeV) and separation using a magnetic field. Potential production routes also include irradiation of rare ^152^Gd targets with high-energy protons (>50 MeV), or spallation reactions using light ions at >500 MeV; however, these methods of production would require mass separation to avoid radioisotopic impurities [36]. ^149^Tb was also produced using the ^nat^Nd(^12^C,xn)^149^Dy -> ^149^Tb method by irradiating a thick target of ^nat^Nd_2_O_3_ with ^12^C ions at 108 MeV. However, the scarcity of ^12^C accelerators limits this production avenue. Another method of production includes the ^151^Eu(^3^He,5n)^149^Tb reaction, where ^151^Eu is bombarded by ^3^He particles from 40–70 MeV. This method results in high yield production of ^149^Tb (3 GBq for an 8 h irradiation), and is advantageous due to target material availability and relatively simple radiochemical processing. The main drawback is the limited availability of high intensity ^3^He beams [25].

### 4.6. Alpha Emitter Handling 

Once produced, α-emitters require special handling and equipment beyond the current capabilities of many radiopharmaceutical centers. Since the external protective layer of skin stops α-particles, pure α-emitters are not an external radiation hazard. However, ingesting and internalizing α-emitters can cause serious effects, including cancer, genetic diseases, teratogenesis, and degenerative changes. To handle α-emitters, specialized equipment to detect α-particles should be employed, such as ZnS(Ag) scintillators to complement Geiger–Muller counters [37]. Detecting α-particles is difficult and time consuming due to the short range of α-particles in air, making a method of long-range detection of α-particles highly desirable [38]. This can be achieved by observing the secondary effects of α-emitters, such as the air-radioluminescence caused by ionization, where α-decay yields up to 400 UV photons in air. These photons have a much longer range in air than α-particles, and in areas with low background UV, they can be used to detect α-particles. One such detector under development has the capability to detect α-particles from a distance of about 40 cm [39]. Due to their properties, handling α-emitters requires much lower contamination removal levels than β^−^-emitting radionuclides [1].

For α-emitters with low energy γ emissions, a well-ventilated fume hood with a glove box is sufficient. For α-emitters with high-energy γ emissions, work should be performed behind 15 cm lead bricks or inside a shielded hot cell with remote manipulator arms [40]. When volatile α-emitters are involved, gas-tight enclosures should be considered, and with all α-emitters, double gloving and γ-counter and liquid scintillation should be employed to prevent and monitor contamination [1]. Additionally, radiopharmacies should have a dedicated space for the clinical production of α-emitters, and dedicated α-emitter waste storage.

## 5. Alpha Emitter Radiochemistry and Targeting

### 5.1. Radiochemistry and Chelators

Chelating agents that result in the stable coordination of α-emitting radionuclides are important, since a matching radiometal–chelate chemistry is key to success in targeted therapies and avoiding unintended distribution and toxicity to nontarget organs and tissues. Metallic radionuclides have utilized bifunctional chelating agents, which have a metal-binding moiety function that sequesters the metallic radionuclide, and a chemically reactive functional group that covalently attaches a targeting vector such as small molecule peptides, proteins, or nanoparticles. The chelator vector can be attached to the vector either directly or through a linker that is often used to modify pharmacokinetics, which can be a simple hydrocarbon chain or a small peptide sequence. The loss of dissociation of the radiometal is associated with therapeutic toxicity; thus, key coordination chemistry factors to consider include charge, similar ionic radius and chelating cavity size, as well as providing an optimal number of chemically appropriate donor binding groups. [41].

From this, the stability of radiopharmaceuticals for use in α-therapy depends on different characteristics, including the coordination properties of the parent radionuclide, complexation kinetics including time, temperature, pH, the thermodynamic stability of the radionuclide–chelator complex in solution, and the kinetic inertness when competing with other ions and chelating agents. Additional factors that influence radiopharmaceutical quality include radiolytic effects, the recoil effects of the daughters, and the unique chemistry of the various daughter nuclides. Several commonly employed chelators are depicted in Figure 3, and notable chelators that have been investigated with therapeutic α emitters are listed in Table 4.

#### 5.1.1. ^225^Ac

In aqueous solution, ^225^Ac is most stable in its +3 oxidation state, with a potential possibility of accessing its +2 oxidation state [48]. Notably, Ac^3+^ is the largest +3 ion on the periodic table, and with a low charge density, it is the most basic +3 ion. Ac^3+^ possesses similar chemical properties to lanthanide Ln^3+^ ions, with lanthanum La^3+^ being a useful Ac^3+^ surrogate for radiochemistry. With its demonstrated ability to stably coordinate hard +3 ions, ^225^Ac is stable on coordination by the DOTA chelator and its derivatives [49], and has shown improvements in whole body clearance and decreased organ uptake compared to acyclic ligands such as EDTA and CHX-A” DTPA. Other ligands, such as macropa and macropa-NCS have promise in coordinating ^225^Ac, and they showed minimal organ accumulation and remained highly stable in vivo over an extended period of time [50,51]. It has also been suggested that acyclic chelating agents such as linear polyaminocarboxylates are not good ligands for ^225^Ac due to high liver uptake and poor whole body clearance resulting from the loss of ^225^Ac from these chelating agents [52]. DOTP^8-^ has been chelated with ^225^Ac, compared with other 3+ cations, including Am, Cm, and La, and verified to be encapsulated within the DOTP^8-^ binding pocket [44]. Recently developed ^225^Ac chelators such as H_4_py4pa and H_4_py4pa-phenyl-NCS have demonstrated excellent in vivo stability and tumor specificity [53]. Another novel chelator, ^225^Ac-crown, was shown to be stable over an extended period of time, while binding rapidly to ^225^Ac at ambient temperature [45].

#### 5.1.2. ^213^Bi

^213^Bi is obtained in its +3 oxidation state from a ^225^Ac/^213^Bi generator, and forms stable complexes with nitrogen-rich chelators such as CHX-A”-DTPA or NETA, and is also stable with DOTA [49]. Optimized protocols for ^213^Bi labelling of antibodies in clinical settings permit the synthesis, sterile filtration, and quality control of therapeutic doses within 15 min of elution from a ^225^Ac generator [32]. It has also been found that kinetically inert Bi^3+^ complex formation is very slow [54]. Notably, phosphorus containing ligands DOTP, DOTP^H^, DOTP^Et^, DOTPI were found to match CHX-A”-DTPA and have superior labelling efficiencies to DOTA, with DOTP being the most efficient. They also exhibited excellent stability in human plasma and exhibited a higher stability against demetallation compared to DOTA and CHX-A”-DTPA [46].

#### 5.1.3. ^212^Bi/^212^Pb

^212^Bi has been attached to a variety of acyclic chelators, notably DTPA, EDTA, EDTMP. The parent nuclide ^212^Pb can be considered preferable to ^212^Bi since its half-life of 10.64 h is significantly longer than the 60.6 min half-life of ^212^Bi, permitting up to 10 times more dose per administered activity. Therefore, ligands that stably chelate ^212^Bi and ^212^Pb are attractive for employing this nanogenerator approach [54].

#### 5.1.4. ^227^Th/^223^Ra

^227^Th has a +4 oxidation state, and has been studied with ligands that accumulate in bone, including phosphate ligands such as DTMP, DOTMP, and EDTMP. Employing ^227^Th is advantageous for obtaining an additional α-decay, compared to just using its ^223^Ra daughter [28]. Although ^223^Ra has a +2 oxidation state, it is highly basic and does not form complexes in aqueous solution. There are no known ^223^Ra chelators, so there is no way to target its accumulation. ^227^Th can also be chelated by hydroxopyridinone coordinating moieties, including N-methyl-3,2-hydroxypyridinone (Me-3,2-HOPO) [55].

#### 5.1.5. ^211^At

^211^At is a halogen, presenting different radiolabeling chemistry than radiometals. In its positive oxidation states, astatine exhibits properties specific to metal ions such as silver in its +1 oxidation state and polonium in its higher oxidation states. In its −1 state, it exhibits characteristics similar to iodine. Slow, low yield electrophilic radiolabeling of nonactivated aromatic rings can lead to ^211^At compounds in vivo [28]. Several compounds that have been stable when labeled with ^211^At include N-succinimidyl 3-(trimethylstannyl)benzoate and a boron cage compound [56,57]. ^211^At can also be radiolabeled by adapting radioiodination chemistry using tin precursors and prosthetic groups [58].

#### 5.1.6. ^149^Tb

^149^Tb forms +3 and +4 oxidation states and can be chelated with DOTA conjugates [59], and it is particularly attractive since there are no α-emitting progeny presenting a redistribution issue upon release from the chelator. Radiolabeling using bifunctional chelators such as DOTA and DTPA has been well developed, giving a clear advantage over alpha emitters such as ^211^At and ^223^Ra [25]. One study using [^149^Tb]Tb-DOTANOC achieved high-quality PET images of an AR42J tumor-bearing mouse, demonstrating the exceptional potential of ^149^Tb to combine α-therapy with PET in “alpha-PET” using a single radionuclide [59].

### 5.2. α Emitter Redistribution

While α emitters with multiple daughters—in vivo generators—can enhance the delivered dose to a tumor site, they can also redistribute and cause unintended toxic effects. Upon α emission, recoil energy (~100 keV) imparted to a daughter nuclide is at least 1000 times higher than any chemical bond, releasing in at least partial release of the daughter nuclide from the targeting molecule. Redistribution depends on the distance traversed by the daughter nuclide upon release, diffusion and active transport processes, the half-life of the daughter, and the affinity of the radionuclide for different organs [1]. With redistribution, radioactive burden can be spread across the body, reducing elimination and leading to radiotoxic effects such as organ dysfunction and secondary tumorigenesis. Dosimetry is essential to understand the contributions of a radiolabeled targeting vector, labeled metabolites, liberated mother nuclides, and the daughters released upon recoil [19]. Radionuclide distribution is often measured in postmortem ex vivo organ analysis, using an alpha camera or a timepix detector [19,60,61,62].

Some α-emitters are prone to redistribution. Redistribution compromised the continuation of a ^224^Ra clinical study, with 8% of the ^220^Rn daughter nuclide leaving the body and significant ^212^Pb and ^212^Bi uptake observed in the red blood cells, kidney, and liver [63]. ^225^Ac faces limitations due to the redistribution of the ^213^Bi daughter to the kidneys, with one study in mice showing 0.77 Gy·kBq^−1^ of kidney dose after tissue harvest, 60% of which was attributed to nonequilibrium ^213^Bi [64]. Dose-limiting salivary gland toxicity and reduced salivary gland function have also been observed in clinical studies using ^225^Ac-labelled PSMA for treating metastatic castration-resistant prostate cancer [65]. Other α emitters such as ^223^Ra demonstrated low redistribution in mice and humans [66]. ^149^Tb shows promise compared to the previously listed α-emitters since it has only one α emission in its decay chain; thus, once ^149^Tb radiopharmaceuticals accumulate in their target, they should be less prone to toxic redistribution effects.

Several theories have been proposed to mitigate the consequences of the daughter recoil effect [19]: (1) Since the spread of daughter radionuclides takes time in the body, their spread depends on their physical half-life. With blood flow in capillaries between 1–3 mm/s, daughter nuclei will not undergo significant translocation if their half-lives are short (several seconds). (2) Daughter recoil can be mitigated by nanoconstructs. Encapsulating a radionuclide within the core of a nanoconstruct with sufficient stopping power can significantly mitigate free daughter spread. (3) Even if a recoil daughter escapes a nanoconstruct, it has a high probability of back-implantation into surrounding nanoconstructs [19].

While α emitters with a shorter t_1/2_ are an effective solution to daughter redistribution, the higher cytotoxicity—and therefore therapeutic potential—of radioisotopes with progeny redistribution has motivated developing techniques to control the daughter radionuclides [1]. Nanocarrier encapsulation of α-emitters to contain recoil daughters shows promise, with one method utilizing liposomes. ^223^Ra was encapsulated in pegylated liposomal doxorubicin (PLD) and remained relatively stable in vivo with skeletal uptake lower than free ^223^Ra [67]. Liposomes have also enhanced the retention of ^225^Ac daughters [68]. One study used TiO_2_ nanoparticles labeled as a carrier for ^225^Ac and functionalized with substance P (5–11), a peptide fragment targeting NK1 glioma cell receptors. Leaching of 30% of ^221^Fr, the first decay daughter of ^225^Ac, was observed in cerebrospinal fluid after 10 days, with the complex showing high cytotoxic effects in T98G glioma cells [69]. One study employed ^225^Ac gold-coated lanthanide phosphate nanoparticles, where the multishell nanoparticles combined the radiation resistance of lanthanide phosphate, magnetic properties of gadolinium phosphate, and gold chemistry for attaching targeting vectors [70]. Another approach involves diffusing alpha-emitters radiation therapy (DaRT), a new form of brachytherapy, where seeds impregnated with radionuclides are embedded in solid tumor tissue where they continually release α-emitters [71].

An additional method to reduce toxic renal effects of ^225^Ac daughters involves metal chelation therapy and diuretics. In animals, oral metal chelation with dithiols was shown to reduce renal ^213^Bi activity, and furosemide and chlorothiazide significantly reduced ^221^Fr renal activity [72].

Pretargeting also shows potential for both PET imaging and for reducing nonspecific toxicity and hematotoxicity in radioimmunotherapy. This is where an unlabeled immunoconjugate capable of binding a tumor specific antigen is injected prior to a small molecular weight payload that then binds to the immunoconjugate [73].

## 6. Preclinical Studies and Clinical Applications

There are numerous preclinical and clinical studies of α-emitting radiopharmaceuticals completed or underway for a variety of cancers, which are outlined in Table 5.

### 6.1. Preclinical Studies

Since α-emitters have the potential for high efficacy but also high toxicity, preclinical studies, in vivo and in vitro, are essential for optimizing α-therapy and guiding further clinical trials. Some notable preclinical studies using a variety of α-emitters are outlined below.

#### 6.1.1. ^225^Ac

Figure 4 shows MRI images of [^225^Ac]Ac-E4G10-treated glioblastoma-bearing mice, demonstrating the clear tumor growth control effected by the treatment. Metastatic prostate cancer is of interest for TAT due to the external domain of the prostate-specific membrane antigen (PSMA). ^225^Ac has been used with PSMA compounds such as PSMA-617 for prostate cancer therapy. One study in PC-3/PC-3-PIP-tumor-bearing mice compared the effects of [^177^Lu]Lu-PSMA617 with [^225^Ac]Ac-PSMA617, finding improved overall antitumor effectiveness and enhanced therapeutic efficacy for [^225^Ac]Ac-PSMA-617 [79]. CD-45 antigens are found on all immune cells, including hematopoietic stem cells and precursor and mature lymphoid and myeloid cells, giving potential for radiotherapy of leukemia and lymphoma. Anti-CD45 radioconjugates labeled with ^225^Ac, such as [^225^Ac]Ac-BC8, have demonstrated effective tumor control in mice bearing multiple myeloma tumors, with modest uptake in the kidney and significant uptake in the liver [77]. Human epidermal growth factor receptor type 2 (HER2) is overexpressed in carcinomas; thus, HER2 antibodies and nanobodies have been investigated for radioimmunotherapy. ^225^Ac-labeled nanobodies such as [^225^Ac]Ac-DOTA-Nb have shown fast uptake in tumor-bearing mice with HER2-overexpressing tumors, with coinjection of Gelofusine reducing kidney retention by 70% [95]. TAT can also target the tumor microenvironment, such as the vasculature and neovascular endothelium. One study using a [^225^Ac]Ac-E4G10 antibody conjugate that targets the vascular endothelium of glioblastoma demonstrated tumor growth control and improved survival in a mouse [81].

#### 6.1.2. ^213^Bi

Figure 5 depicts the survival curve of mice treated with ^213^Bi-9E7.4 anti-CD138 antibodies compared to a control group. ^213^Bi has been shown to be more effective than ^177^Lu in animal models for treatment of multiple myeloma (MM). Using ^213^Bi-labeled 9E7.4 anti-CD138 antibodies in mice with 5T33 MM cells, mean survival was significantly increased significantly and a cure was effected in 45% of animals. ^177^Lu-labeled 9E7.4 anti-CD138 antibodies increased survival to a lesser degree than the ^213^Bi, with no mice cured [96]. [^213^Bi]Bi-DOTA-biotin has been used to target non-Hodgkin lymphoma when pretargeted with an anti-CD20 fusion protein. Mice with Ramos lymphoma xenografts exhibited significant delayed tumor growth and no treatment-related mortalities [84]. Another study in mice with intramuscular LNCaP xenografts used ^213^Bi conjugated to a J591 anti-PSMA monoclonal antibody, which resulted in improved tumor-free survival, effectively stopped the growth of LNCaP spheroids, and reduced prostate-specific antigen levels [97].

#### 6.1.3. ^223^Ra.

The antitumor effects of ^223^Ra have been demonstrated extensively in animal models, leading to commercial use of this radium isotope in men with metastatic prostate cancer, where it has improved overall survival and reduced the time until the first symptomatic skeletal event [98]. ^223^Ra was investigated in an experimental skeletal metastases model in nude rats that received human breast cancer cells. The animals treated with ^223^Ra demonstrated significant antitumor effects and improved survival rates, with no signs of bone marrow toxicity or body weight loss. This study led to further development of ^223^Ra as a bone-marrow-sparing treatment for skeletal cancers [88].

#### 6.1.4. ^224^Ra

^224^Ra is similar to ^223^Ra with its tendency to adsorb to bone at sites of active mineral crystallization, and inability to be stably bound to a targeting molecule vector. Calcium carbonate microparticles have been proposed as ^224^Ra carriers for treatment of disseminated cancers, such as in those occurring in the peritoneum. One study demonstrated high labelling efficiencies of ^224^Ra on the surface of calcium carbonate microparticles with high retention of both ^224^Ra and its ^212^Pb daughter, and the radioactivity remained primarily localized in the peritoneal cavity [89]. ^224^Ra has also been investigated for osteolytic bone metastasis of MDA-MB-231(SA)-GFP-bearing nude mice, decreasing the number and area of tumor foci and prolonging survival [99]. Preclinical diffusing alpha emitters therapy (DaRT) using ^224^Ra-loaded wires was tested on mice bearing prostate (PC-3), glioblastoma (GBM, U87-MG), colon (HCT15), squamous cell carcinoma (FaDu), and melanoma (C32) cancer cell tumors. The in vivo study confirmed that DaRT can destroy tumors, with only the C32 cells exhibiting resistance [100].

#### 6.1.5. ^212^Pb

The ^224^Ra daughter ^212^Pb has been used in a variety of preclinical trials. ^212^Pb-labeled 376.96 monoclonal antibodies were used in mice bearing pancreatic cancer Panc039 xenografts, exhibiting significant uptake and tumor growth inhibition compared to an existing [^212^Pb]Pb-F3-C25 compound [90]. ^212^Pb-labeled 225.28 antibodies were used in immune-deficient mice bearing SUM159 and 2LMP human triple-negative breast cancer cells, showing high cell uptake and effective inhibition of tumor growth for CSPG4-expressing tumors [78]. Recently, a variety of low molecular weight ligands were developed and labeled with ^212^Pb, and investigated in PSMA(+) PC3 PIP and PSMA(−) PC3 flu flank xenografts. Several of the labeled ligands demonstrated antitumor efficacy, with the kidneys discovered to be the dose-limiting organ [101]. Internalizing and non-internalizing antibodies can have different outcomes. Using internalizing ^212^Pb-labeled trastuzumab and non-internalizing [^212^Pb]Pb-35A7 in mice with intraperitoneal A-431 cell tumor xenografts, the non-internalizing conjugate led to higher tumor dose, while the internalizing led to longer mean survival. This demonstrates the potential advantage of using internalizing TAT [102]. ^212^Pb-labeled NG001, a PSMA ligand, was investigated in mice bearing C4-2 tumors, demonstrating a 2.5-fold lower kidney uptake compared to [^212^Pb]Pb-PSMA-617. This was attributed to the NG001 chelator having all four arms available due to the use of a backbone linker, instead of using a chelator arm for the linking purpose. [103].

#### 6.1.6. ^227^Th

^227^Th has been conjugated to antibodies including trastuzumab, rituximab, and has demonstrated a significant delay in tumor growth and prolonged survival in breast, ovarian, and lymphoma models [3]. [^227^Th]Th-rituximab was investigated in mice with human lymphoma Raji xenografts. The maximum tolerated dose was found to be 600–1000 kBq/kg, with a 1000 kBq/kg dose resulting in significant weight loss and temporary drop in white blood cell and platelet counts, with no significant signs of toxicity in examined tissue [85]. Lintuzumab, an anti-CD33 antibody, has been used extensively in TAT for treatment of myeloid leukemia, and has been conjugated to ^213^Bi, ^225^Ac, and ^227^Th. [3]. [^227^Th]Th-lintuzumab induced cytotoxicity in CD33-positive cells and demonstrated antitumor activity in a HL-60 cell mouse model with a single dose regimen [86].

#### 6.1.7. ^211^At

^211^At has been investigated for non-small cell lung cancer using an ^211^At-labeled octreotide somatostatin analogue. In mice, [^211^At]At-SPC-octreotide exhibited higher uptake in the lung, spleen, stomach, and intestines with rapid clearance 24 h post-injection, while demonstrating tumor cell apoptosis in a dose-dependent manner [91]. ^211^At-labeled anti-Frizzled homolog 10 antibody, [^211^At]At-OTSA101, has been shown to suppress the growth of synovial sarcoma xenografts in mice with greater efficiency than β^−^ emitter [^90^Y]Y-OTSA101 [92]. Anti-HER2 single domain antibodies and nanobodies have been evaluated, with iso-[^211^At]At-SAGMB-5F7 demonstrating high intracellular trapping of radioactivity and greater than 10:1 tumor-to-normal organ ratios except for the kidneys and lungs by 2 h post-injection [104]. Another compound, [^211^At]At-SAGMB-2Rs15d, demonstrated high tumor uptake, low background signals, and fast renal excretion [105]. ^211^At has also been conjugated to anti-CD38 and anti-CD45 antibodies. ^211^At-CD38 produced a sustained remission and long-term survival (>150 days) for 50 to 80% of mice with multiple myeloma xenografts, compared to untreated mice dying in 20 to 55 days [106]. [^211^At]At-anti MICA/B antibodies have shown significant reduction in the tumor growth rate of HCT116 p53^−/−^ xenografts in mice. Systemic cancers such as B-cell lymphoma can be treated by targeting the CD20 surface antigen [107].6.1.8. ^149^Tb

Figure 6 depicts PET/CT images of an AR42J tumor-bearing mouse after injection with [^149^Tb]Tb-DOTANOC. ^149^Tb was first used in a SCID mouse model of leukemia with ^149^Tb-labeled rituximab. In total, 89% of treated mice experienced tumor-free survival of over 4 months, a significant increase compared to other groups [108]. Since then, the ^149^Tb-labeled DOTA–folate conjugate has been evaluated in folate receptor (FR)-positive cancer in KB-tumor-bearing mice, with an observed dose-dependent effect, significant tumor growth delay, and no signs of acute toxicity to the kidneys or liver [36]. ^149^Tb is also a positron emitter; thus, it was investigated for its utility in PET imaging using [^149^Tb]Tb-DOTANOC in a mouse with AR42J tumor xenografts, producing distinct visualization of the tumors with residual radioactivity in the kidneys and bladder [59]. The potential for ^149^Tb alpha-PET would make it attractive for clinical applications, allowing the direct visualization of therapeutic ^149^Tb-radioligands in vivo [24]. There are currently no listed clinical trials for ^149^Tb; additional preclinical studies and increased supply sources may enable future clinical trials.

### 6.2. Clinical Studies

In the past decade, there has been a large increase in TAT clinical trials using an array of α-emitters for treating a variety of cancers. Notably, many trials listed on clinical-trials.gov are currently underway, recruiting, or are not yet open for recruitment. A summary of completed trials and those underway or about to begin is given in this section.

#### 6.2.1. ^225^Ac

Figure 7 depicts the results from [^225^Ac]Ac-PSMA-617 therapy in a human patient, which overcame tumor resistance to [^177^Lu]Lu-PSMA-617. There are nine clinical trials listed on clinical-trials.gov for ^225^Ac, one of which is a completed phase 1 trial using ^225^Ac-labeled humanized anti-CD33 HuM195 antibodies for advanced myeloid malignancies including leukemia and myelodysplastic syndrome. ^225^Ac was considered as an alternative to ^213^Bi for its greater cytotoxic potential and longer half-life. [^225^Ac]Ac-HuM195 eliminated peripheral blasts in 63% of patients at doses of 37 kBq/kg or higher, and reduced bone marrow blasts in 67% of patients [87]. ^225^Ac salvage therapy overcame β^−^ resistance in two metastatic-CRPC patients after a PSMA-positive tumor phenotype was verified by [^68^Ga]Ga-PSMA-11 PET/CT. Both patients exhibited a complete response, and PSA dropped below the measurable level, with side effects of xerostomia observed in both patients. A further study with 14 metastatic CRPC patients identified 100 kBq/kg to be an appropriate balance between toxicity and antitumor activity [16]. While [^177^Lu]Lu-PSMA-617 has improved overall survival of CRPC patients, it has been suggested that further studies with ^225^Ac PSMA compounds will elucidate their distinct advantages over β^−^ therapy with ^177^Lu [109,110]. A pilot study using [^225^Ac]Ac-PSMA-617 in patients heavily pretreated with chemotherapy led to a greater than 90% decline in PSA serum in 82% of patients, including 41% of patients with undetectable serum PSA who remained in remission 12 months after therapy [111]. A more recent study reported a single cycle of [^225^Ac]Ac-DOTATOC achieving a partial remission in a patient with refractory metastatic neuroendocrine tumors without any adverse effects after resistance to 10 cycles of the β^−^-emitters [^177^Lu]Lu-DOTATATE and [^90^Y]Y-DOTATOC [76]. Ongoing trials with ^225^Ac include a phase 1 study of [^225^Ac]Ac-lintuzumab in patients with refractory multiple myeloma; an early phase 1 trial of [^225^Ac]Ac-PSMA radioligand therapy of metastatic castration-resistant prostate cancer; a phase 1 study that targets type I insulin-like growth factor receptor using [^225^Ac]Ac-FPI-1434 injection for patients with advanced refractory solid tumors; an early phase 1 trial of [^225^Ac]Ac-PSMA radioligand therapy of metastatic castration-resistant prostate cancer; a phase 1 and 2 trial with [^225^Ac]Ac-lintuzumab in older acute myeloid leukemia patients; a phase 1 trial using [^225^Ac]Ac-lintuzumab in combination with CLAG-M chemotherapy in patients with relapsed/refractory acute myeloid leukemia; a phase 1 and 2 trial of venetoclax and [^225^Ac]Ac-lintuzumab in acute and relapsed myeloid leukemia patients; and a phase 1 and 2 trial venetoclax, azacytidine, and [^225^Ac]Ac-lintuzumab in acute myeloid leukemia patients.

#### 6.2.2. ^213^Bi

Figure 8 depicts the significant therapeutic efficacy of [^213^Bi]Bi-DOTATOC on liver metastases in a patient. [^213^Bi]Bi-DOTATOC has been used for β^−^ radiation–refractory tumors in patients with advanced neuroendocrine tumors and liver metastasis who were pretreated with β^−^ emitting [^90^Y]Y-DOTATOC and [^177^Lu]Lu-DOTATOC. [^213^Bi]Bi-DOTATOC was administered with increasing activity in cycles every 2 months, with patients overcoming β^−^ radiation resistance and showing long-lasting antitumor response. Renal toxicity was minimized by administering lysine, arginine, and Gelofusine as developed for β^−^ therapy [75]. Advanced myeloid leukemia has been treated with [^213^Bi]Bi-HuM195, with doses from 10.4–37 MBq/kg. Although patients developed transient myelosuppression, no extramedullary toxicity was observed. However, no patient achieved complete remission, likely due their large tumor burdens. [^213^Bi]Bi-DOTA-substance P has been used in a study with 20 patients with recurrent glioblastoma, with only mild and transient adverse reactions and a median survival after recurrence of 10.9 months [82]. Another study used ^213^Bi-labeled anti-EGFR monoclonal antibodies in patients with carcinoma in situ of the bladder, achieving complete remission in 3 of 12 patients with no adverse effects observed and blood and urine parameters remaining in normal ranges [93].

#### 6.2.3. ^223^Ra

Having received FDA approval in May 2013 to treat castration-resistant prostate cancer under the name Xofigo, ^223^Ra-dichloride has been studied extensively, with 108 listed clinical trials on the clinicaltrials.gov website. ^223^Ra-dichloride resulted in pain relief and a reduction in alkaline phosphatase in the first clinical trial in patients with skeletal metastases. Over 50% of patients reporting pain relief 8 weeks after injection, and median survival exceeded 20 months [112]. One ^223^Ra-dichloride phase III trial with 921 patients found that the time to first symptomatic skeletal event (15.6 months) was significantly longer compared to the placebo group (9.8 months). The risks of external beam radiation therapy for bone pain and spinal cord compression were also found to be reduced [113]. Another study in CRPC patients found overall survival was significantly longer with ^223^Ra-dichloride at 14.9 months compared to 11.3 months with a placebo. ^223^Ra treatment was associated with low myelosuppression and reduced adverse events. [114]. These studies collectively found that ^223^Ra-dichloride is well tolerated and minimally toxic, and patients have reported an increase in quality of life. Despite these benefits, ^223^Ra does not target soft tissue or circulating disease components [3]. However, in 2018, the European Medicines Agency recommended restricting the use of Xofigo to patients who had two previous treatments for metastatic prostate cancer, or who could not receive other treatments. The agency concluded that Xofigo should not be used in combination with Zytiga, prednisone, or prednisolone, since it was observed to reduce survival and lead to additional bone fractures in patients also taking Zytiga [115].

#### 6.2.4. ^224^Ra

Figure 9 depicts a complete response to squamous cell carcinoma of the scalp using DaRT and a Kaplan-Meier plot of progression-free survival stratified by complete and partial response. There are currently eleven ^224^Ra clinical trials listed on clinicaltrials.gov. Nine of these trials are utilizing diffusing alpha radiation emitters therapy for the treatment of skin cancer, mucosal neoplasms of the oral cavity, soft tissue neoplasms, pancreatic cancer, squamous cell carcinoma, and breast cancer. The other two trials are using Radspherin^®^, which consists of a ^224^Ra α-emitting calcium carbonate microsphere for patients with peritoneal carcinoma, colorectal carcinoma, and ovarian cancer. A recent DaRT study used ^224^Ra seeds to treat squamous cancers of the skin and head and evaluated early tumor responses 30 to 45 days post insertion, with complete response to the treatment observed in 22 of 28 patients [94].

#### 6.2.5. ^212^Pb

There are two clinical trials listed for ^212^Pb, with a phase 1 safety study using [^212^Pb]Pb-TCMC-trastuzumab in patients with HER2-expressing ovarian cancer confined to the peritoneal cavity. After intraperitoneal injection, minimal myelosuppression and radiopharmaceutical redistribution outside the peritoneal cavity were observed; however, no patient exhibited a partial response [116]. The other trial is a phase 1 study using AlphaMedix™ ([^212^Pb]Pb-DOTAMTATE) in patients with metastatic somatostatin-receptor-positive neuroendocrine tumors.

#### 6.2.6. ^227^Th

There are four clinical trials using ^227^Th listed. They include a phase 1 trial using the thorium labeled PSMA-TTC (BAY 2315497) immunoconjugate consisting of a human anti-PSMA antibody covalently linked to the chelator moiety (3,2 HOPO) radiolabeled to ^227^Th for use in patients with metastatic castration resistant prostate cancer [80]; a phase 1 trial using the ^227^Th-labeled antibody–chelator conjugate MSLN-TTC (BAY 2287411) in patients with mesothelin expressing tumors; and a phase 1 study using the ^227^Th-labeled antibody BAY2701439 in patients with breast, gastric, gastroesophageal, and other cancers expressing HER2. The fourth, a now complete phase I trial, used [^227^Th]Th-epratuzumab (BAY1862864) in patients with relapsed or refractory CD-22-positive non-Hodgkin lymphoma.

#### 6.2.7. ^211^At

There are multiple ^211^At clinical trials, with a phase I and II trial using [^211^At]At-labeled BC8-B10 monoclonal antibodies for the treatment of nonmalignant diseases in patients undergoing hematopoietic cell transplant, two trials both in phase I and II using [^211^At]At-labeled BC8-B10 antibodies in patients with high-risk myeloid leukemia, acute lymphoblastic leukemia, myelodysplastic syndrome [117], or mixed-phenotype acute leukemia, and a completed trial with 18 patients using the ^211^At-labeled ch81C6 antibody with primary or metastatic brain tumors. In this trial, no patients experienced dose-limiting toxicity, there was no identified maximum tolerated dose, and median survival increased to 54.1 weeks from 23 weeks. Another clinical trial evaluated the absorbed dose and investigated the toxicity of [^211^At]At-MX35 F(ab’)^2^. It targeted the sodium-dependent phosphate transport protein 2b in patients with complete clinical remission after second-line chemotherapy for recurrent ovarian carcinoma. This trial established that intraperitoneal administration of [^211^At]At-MX35 F(ab’)^2^ could achieve therapeutic doses without significant toxicity in microscopic tumor clusters [118].

## 7. Future of Alpha Therapy

The future of α-therapy holds significant promise for therapeutic clinical applications. Key to its ongoing success is the expansion of robust and high-yield production routes to enhance α-emitter availability, the development of new chelators, linkers, and vectors to enhance efficacy and targeting specificity, and finding solutions to progeny redistribution induced toxicity.

With additional α-emitter production, larger scale preclinical and clinical trials with α-emitters should become possible. The construction of new high-energy particle accelerators should also support more preclinical studies and clinical trials of the appealing ^149^Tb α-emitter. This could lead to more widespread use of TAT beyond FDA approved ^223^Ra-dichloride, enabling the targeting of a wide range of soft tissue and circulatory disease components.

Controlling the progeny is a challenge with existing α-emitters. Despite the ability to deliver significant dose through a nanogenerator approach, the cytotoxic effects of α-emitter progeny in the ^225^Ac, ^227^Th, ^224^Ra, and ^211^At decay chains have shown that balancing tolerable dose with cytotoxic effects can be difficult to achieve. Some solutions to control the daughters or hasten their excretion have included radionuclide impregnated wires, cellular internalization, nanocarriers, metal chelation therapy, and diuretics. Alternatively, the absence of α-emitting progeny in the ^149^Tb decay chain could warrant its preference over currently employed α-emitters once its availability improves.

TAT would also benefit from more investigation into dosimetry and radiobiology. Given the majority of DNA lesions along an α-particle path are double strand breaks, this differentiates α-therapy from other classes of therapeutic radionuclides that primarily cause single strand breaks and can be rendered ineffective due to cellular adaptative and resistance mechanisms. Determining an effective microdosimetry method and improving the understanding of α-emitter radiobiology is important for optimizing their efficacy and safe clinical implementation.

## 8. Conclusions

The short-range high LET characteristics of α-emitter radiotherapy hold promise in being an effective therapy for a variety of cancers. The highly cytotoxic DNA double-strand breaks and secondary cross-dose and bystander effects give α-emitters a formidable advantage over other forms of radiotherapy, including β^−^ or auger electron therapy. Supported by ongoing in vitro and in vivo advances in radiochemistry and efficacy in preclinical models, existing clinical trials have demonstrated its feasibility in patients with many additional clinical trials underway or scheduled to start soon. To date, α-emitting isotopes have been delivered in free form, with targeting vectors or carriers designed to target specific receptors on cells and limit the spread of the daughters, and the more recent DaRT approach of implantable controlled release. Despite uncertainties in α-therapy regarding optimal patient dose and unintended organ toxicity, preclinical animal models have demonstrated marked improvements. Further advances in production and availability, along with management of the radionuclide progeny, should allow for the development and more cost-effective clinical adoption of TAT. There is immense potential for TAT to complement existing forms of therapy and improve the treatment options and quality of life of patients with highly resistant and late-stage diseases.

## Figures and Tables

**Figure 1 pharmaceutics-13-00049-f001:**
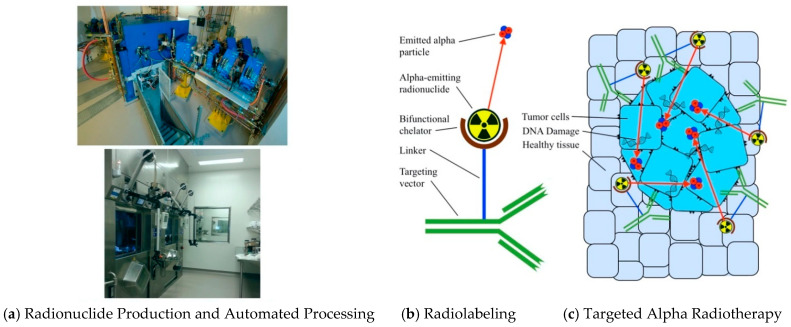
Key aspects of targeted alpha therapy: (**a**) radionuclide production via cyclotron, nuclear reactor or generator decay, and shielded automated processing; (**b**) radiolabeling the alpha-emitting radionuclide to a suitable targeting vector to form a bioconjugate; and (**c**) targeted alpha radiotherapy precisely destroys tumor cells while sparing surrounding healthy tissue.

**Figure 2 pharmaceutics-13-00049-f002:**
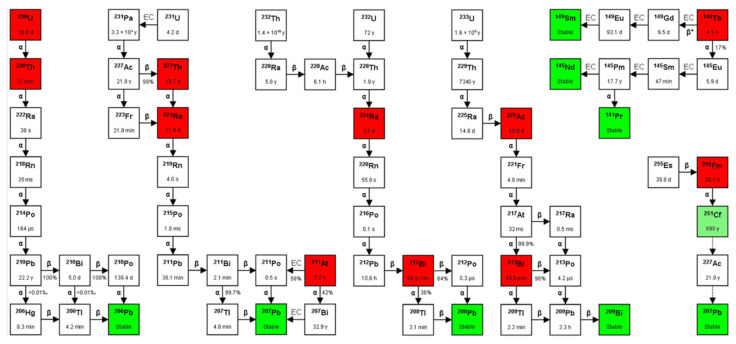
Decay chains of some common therapeutic α-emitters. Reproduced from [6], Frontiers, 2014.

**Figure 3 pharmaceutics-13-00049-f003:**
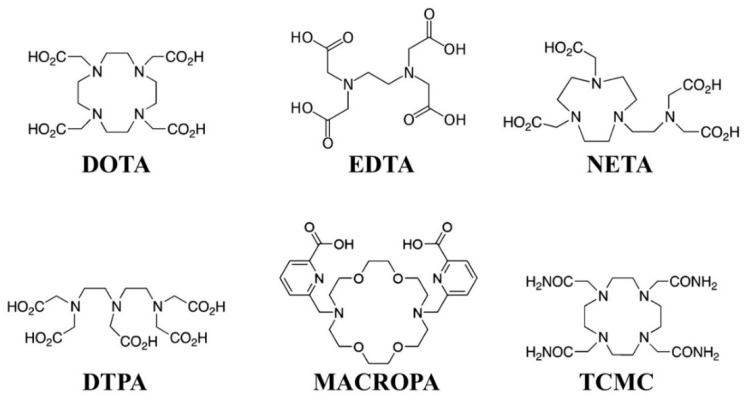
Several commonly employed radiometal chelators.

**Figure 4 pharmaceutics-13-00049-f004:**
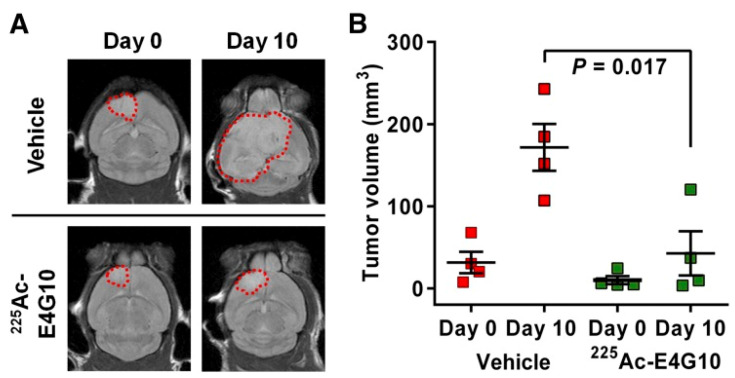
(**A**) Tumor sizes of control and [^225^Ac]Ac-E4G10 treated glioblastoma-bearing mice before and after 10 days, imaged by MRI. (**B**) Mean tumor volumes for control and [^225^Ac]Ac-E4G10-treated mice. Reproduced from [81], JNM, 2016.

**Figure 5 pharmaceutics-13-00049-f005:**
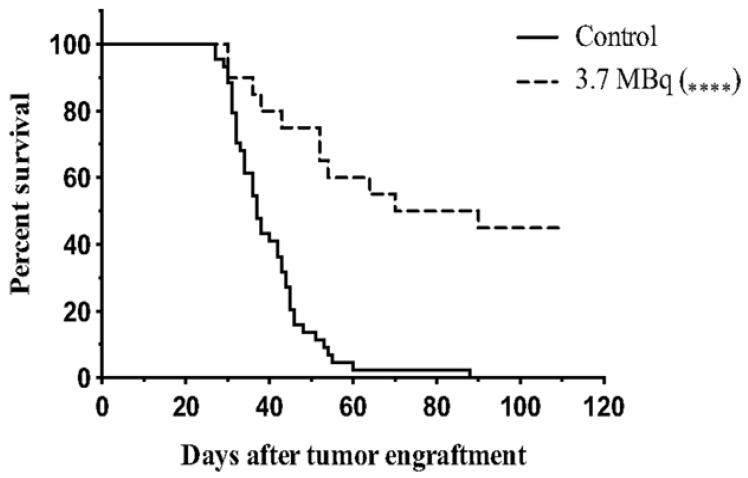
Survival curve of mice injected with 5T33 cells at day 0. 3.7 MBq of ^213^Bi-9E7.4 antibodies was injected for alpha therapy (*n* = 20), and NaCl was injected into the control group (*n* = 44). Reproduced from [96], Frontiers, 2015.

**Figure 6 pharmaceutics-13-00049-f006:**
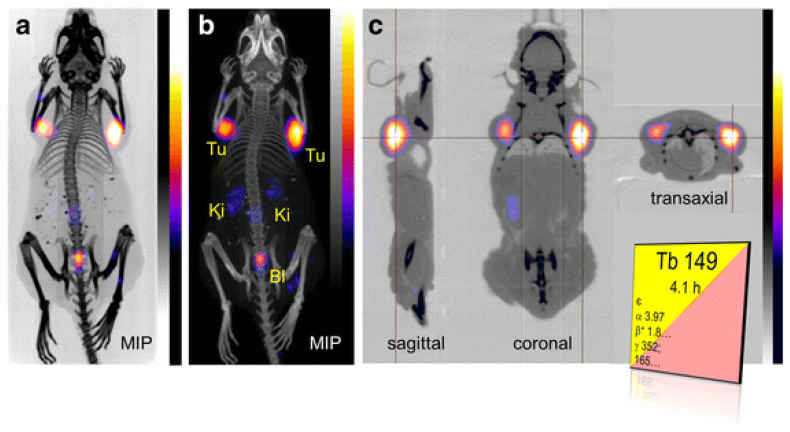
(**A**,**B**) Maximal intensity projections and (**C**) sections of positron emission tomography (PET)/CT images of an AR42J tumor-bearing mouse 2h after injection with 7 MBq of [^149^Tb]Tb-DOTANOC. Reproduced from [59], Frontiers, 2017.

**Figure 7 pharmaceutics-13-00049-f007:**
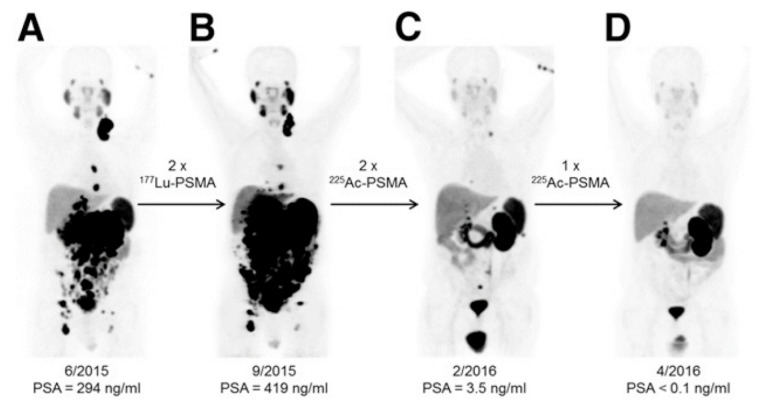
[^68^Ga]Ga-PSMA-11 PET/CT scans of a patient with castration-resistant prostate cancer (CRPC). (**A**) Initial tumor burden (**B**) Progression despite 2 cycles of β^−^-emitting [^177^Lu]Lu-PSMA-617 (**C**,**D**) Impressive decrease in tumor burden after two cycles of α-emitting [^225^Ac]Ac-PSMA-617. Reproduced from [6], Frontiers, 2014.

**Figure 8 pharmaceutics-13-00049-f008:**
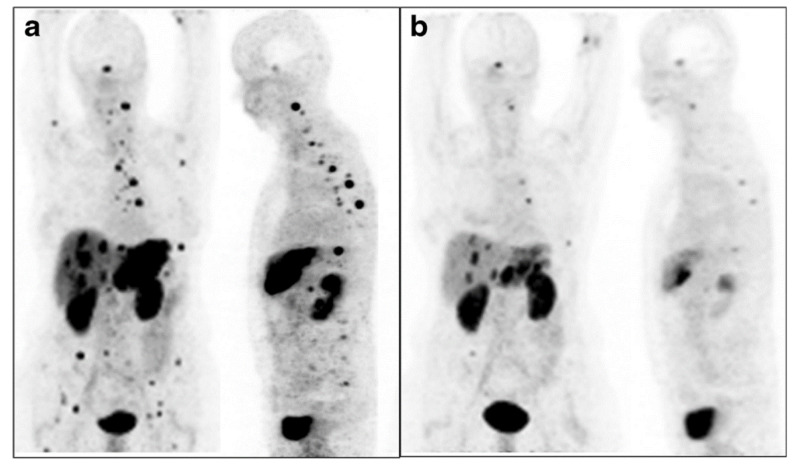
(**a**) Patient with an extensive liver metastases tumor burden imaged with [^68^Ga]Ga-DOTATOC. (**b**) After injection of 10.5 GBq of [^213^Bi]Bi-DOTATOC, liver metastases shrunk significantly. Reproduced from [75], Springer, 2014.

**Figure 9 pharmaceutics-13-00049-f009:**
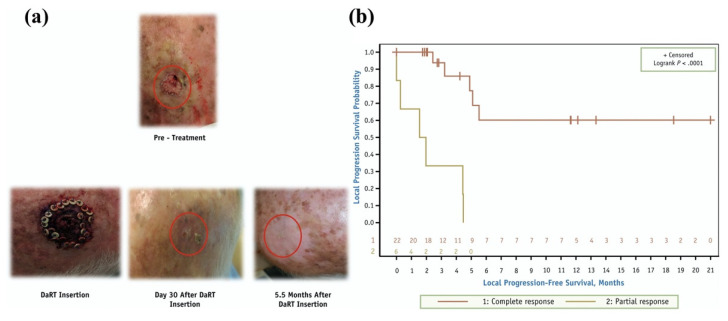
(**a**) Deeply infiltrating squamous cell carcinoma of the scalp showing complete response to diffusing alpha emitters therapy (DaRT) by day 30. (**b**) Kaplan-Meier local progression-free survival stratified by complete and partial response. Reproduced with permission from Popovtzer, A., published by Elsevier, 2020 [94].

**Table 1 pharmaceutics-13-00049-t001:** Key characteristics of α, β^−^, and auger electron emitters and their clinical applications.

Radioactive Particle	Decay Characteristics	Clinical Cancer Applications	Reference
Beta particle(β^−^)	Emission energy per decay: 50–2300 keV Range: 0.05–12 mm Linear Energy Transfer (LET): 0.2 keV/µm	Metastatic castration resistant prostate cancer, acute myeloid leukemia, neuroendocrine tumors, acute lymphocytic leukemia, ovarian carcinomas, gliomas, metastatic melanoma, colon cancer, bone metastases	[1,3,4]
Auger electron (AE)	Emission energy per decay: 0.2–200 keV Range: 2–500 nm LET: 4–26 keV/µm	Advanced pancreatic cancer with resistant neoplastic meningitis, advanced sst-2 positive neuroendocrine and liver malignancies, metastatic epidermal growth factor receptor (EGFR)-positive breast cancer, glioblastoma multiforme	[1,5]
Alpha particle (α)	Emission energy per decay: 5–9 MeV Range: 40–100 µm LET: 80 keV/µm	Metastatic castration resistant prostate cancer, relapsed or refractory CD-22-positive non-Hodgkin lymphoma, acute myeloid leukemia, neuroendocrine tumors, ovarian carcinoma, gliomas, intralesional and systemic melanoma, colon cancer, bone metastases	[1,3,4]

**Table 2 pharmaceutics-13-00049-t002:** Notable α-emitters and their daughters, half-lives, decay energies, and emission types [20].

Parent	Daughters	Half-Life	Emission Type (Energy, Intensity)				
			α	β^−^	β^+^	γ	X-ray
^225^Ac		9.9 d	5.8 MeV, 50.7%			100 keV, 1%	18.6 keV, 13%
	^221^Fr	4.8 min	6.3 MeV, 83.3%			218 keV, 11.4%	17.5 keV, 2%
	^217^At	32.3 ms	7.1 MeV, 99.9%				
	^213^Bi	45.6 min	5.9 MeV, 1.9%	492 keV, 66%		440 keV, 26%	79 keV, 1.8%
	^213^Po	3.72 μs	8.4 MeV, 100%				
	^209^Tl	2.16 min		178 keV, 0.4%		1567 keV, 99.7%	75 keV, 9.7%
	^209^Pb	3.23 h		198 keV, 100%			
	^209^Bi	Stable					

^224^Ra		3.63 d	5.7 MeV, 95%			241 keV, 4.1%	
	^220^Rn	55.6 s	6.3 MeV, 99.9%				
	^216^Po	0.15 s	6.8 MeV, 99.9%				
	^212^Pb	10.6 h		93.5 keV, 83%		238 keV, 43.6%	77 keV, 17.5%
	^212^Bi	60.6 min	6.1 MeV, 25%	834 keV, 55%		727 keV, 6.7%	15 keV, 7%
	^212^Po	0.30 μs	8.8 MeV, 100%				
	^208^Tl	3.1 min		650 keV, 49%		2614 keV, 99.9%	
	^208^Pb	Stable					

^227^Th		18.7 d	6.0 MeV, 100%			236 keV, 13%	19 keV, 37%
	^223^Ra	11.4 d	5.7 MeV, 100%			269 keV, 14%	83 keV, 25%
	^219^Rn	3.96 s	6.8 MeV, 79.4%			271 keV, 10%	16 keV, 1%
	^215^Po	1.78 ms	7.4 MeV, 99.9%				
	^211^Pb	36.1 min		471 keV, 91%		404 keV, 3.8%	
	^211^Bi	2.14 min	6.6 MeV, 83.5%	172 keV, 0.3%		351 keV, 13%	
	^207^Tl	4.77 min		492 keV, 99.7%			
	^207^Pb	Stable					

^211^At		7.2 h	5.9 MeV, 42%				79 keV, 21%
	^211^Po	0.52 s	7.5 MeV, 98.9%				
	^207^Bi	31.6 y				570 keV, 97.8%	
	^207^Pb	Stable					

^149^Tb		4.1 h	4.0 MeV, 16.7%		638 keV, 3.8%	352 keV, 29.4%	43 keV, 36%
	^149^Gd	9.3 d				150 keV, 48%	42 keV, 55%
	^149^Eu	93.1 d					40 keV, 40%
	^149^Sm	Stable					
	^145^Eu	5.9 d			740 keV, 1.5%	894 keV, 66%	40 keV, 40%
	^145^Sm	340.3 d				61 keV, 12%	39 keV, 71%
	^145^Pm	17.7 y				72 keV, 2%	37 keV, 40%
	^145^Nd	Stable					

**Table 3 pharmaceutics-13-00049-t003:** Current and anticipated production methods for therapeutic alpha emitter systems.

α Emitter	Production Method	Status	Reference
^225^Ac	^229^Th/^225^Ac generator	Production	[26]
	^226^Ra(p,2n)^225^Ac	Research	[26]
	^226^Ra(γ, n)^225^Ra	Potential	[26]
	^226^Ra(n,2n)^225^Ra	Potential	[26]
	^226^Ra(d,3n)^225^Ac	Potential	[26]
	^232^Th(p,x)^225^Ac	Research	[26]
^213^Bi	^225^Ac generator	Production	[17]
^224^Ra	^228^Th/^224^Ra generator	Previously used	[27]
^212^Bi	^224^Ra/^212^Bi generator	Production	[28]
^212^Pb	^224^Ra/^212^Pb generator	Production	[28]
^227^Th	^227^Ac decay	Production	[28]
	^235^U decay	Production	[28]
^223^Ra	^227^Th/^223^Ra generator	Production	[28]
^211^At	^209^Bi(α,2n)^211^At	Production	[29]
	^232^Th(p,x)^211^Rn	Research	[28]
	^238^U(p,x)^211^Rn	Research	[28]
	^209^Bi(^7^Li,5n)^211^Rn	Research	[30]
	^209^Bi(^6^Li,4n)^211^Rn	Research	[28]
^149^Tb	^152^Gd(p,4n)^149^Tb	Research	[25]
	^nat^Nd(^12^C,xn)^149^Dy -> ^149^Tb	Research	[31]
	^151^Eu(^3^He,5n)^149^Tb	Research	[25]
	^nat^Ta(p,x)^149^Tb	Research	[25]
	^141^Pr(^12^C,4n)^149^Tb	Research	[31]

**Table 4 pharmaceutics-13-00049-t004:** Chelators for therapeutic α emitters.

Radionuclide	Commonly Investigated Chelators	References
^225^Ac	DOTA, DOTATOC, DO3A, PEPA, EDTA, CHX-A”-DTPA, HEHA, DOTMP, tu-BU-calix [4]arene-tetracarboxylic acid, macropa, macropa-NCS, H_4_py4pa, H_4_octapa, H_4_*CHX*octapa, DOTP, crown	[21,26,28,42,43,44,45]
^213^Bi	DOTA, DOTATOC, DTPA, CHX-A”-DTPA, DOTP, DOTPH, DOTPOEt, DOTPI, NETA	[28,46]
^212^Bi	DOTA, DOTMP, DTPA, TCMC, 1B4M-DTPA, CHX-A”-DTPA, NETA	[28,47]
^212^Pb	DOTA, DTPA, TCMC, EDTA	[27,34,40]
^227^Th	DOTA, DTPA, DTMP, DOTMP, HOPO, octapa me-3,2-HOPO	[28]
^211^At	m-or p-SnMe_3_-Bz, m-or p-SnBu_3_-Bz, closo-decaborate, Tin precursors, prosthetic groups	[1,28]
^223^Ra	No known chelators	[28]
^224^Ra	No known chelators	[28]
^149^Tb	DOTA, DOTANOC, DOTA-folate, DTPA	[36]

**Table 5 pharmaceutics-13-00049-t005:** Preclinical investigations and clinical applications of α-emitting radiopharmaceuticals.

Cancer Type	α-Emitting Radiopharmaceutical		Reference
	Preclinical	Clinical	
Colorectal cancer	^213^Bi-labeled CO-1A Fab’, ^224^Ra diffusing alpha emitters radiation therapy (DaRT)	^224^Ra (Radspherin^®^)	[74]
Neuroendocrine	[^225^Ac]Ac-DOTATOC	[^213^Bi]Bi-DOTATOC, [^212^Pb]Pb-DOTAMTATE, [^225^Ac]Ac-DOTATOC	[74,75,76]
Multiple myeloma	^213^Bi-labeled 9.E7.4 anti-CD138 mAb, [^225^Ac]Ac-BC8, [^211^At]At-CD38	[^225^Ac]Ac-lintuzumab	[74,77]
Breast cancer	[^225^Ac]Ac-7.16.4 anti-rat HER-2/neu, [^212^Pb]Pb-labeled 225.28 antibodies,	^224^Ra (DaRT), ^227^Th-antibody	[74,78]
Metastatic castration resistant prostate cancer	[^213^Bi]Bi-DOTA-PESIN	[^225^Ac]Ac-PSMA617, [^223^Ra]Ra-dichloride (Xofigo), [^227^Th]Th-PSMA antibody	[74,79,80]
Peritoneal carcinoma	[^213^Bi]Bi-d9MAb	^224^Ra (Radspherin^®^)	[74]
Glioblastoma	[^225^Ac]Ac-E4G10	[^213^Bi]Bi-DOTA-substance P, [^211^At]At-ch81C6	[81,82,83]
Lymphoma	[^213^Bi]Bi-DOTA-biotin, [^227^Th]Th-rituximab, [^227^Th]Th-epratuzumab		[84,85]
Leukemia	[^213^Bi]Bi-lintuzumab, [^227^Th]Th-lintuzumab, [^149^Tb]Tb-rituximab	[^225^Ac]Ac-anti-CD33 HUM195, [^225^Ac]Ac-lintuzumab, [^213^Bi]Bi-HuM195, [^211^At]At-BC8-B10	[3,86,87]
Skeletal cancers and bone metastases		^223^Ra, ^224^Ra	[88,89]
Ovarian cancer		^224^Ra (Radspherin^®^), [^212^Pb]Pb-TCMC-trastuzumab	
Pancreatic cancer	^212^Pb-labeled 376.96 mAb	^224^Ra (DaRT),	[90]
Lung cancer	[^211^At]At-SPC-octerotide		[91]
Synovial Sarcoma	[^211^At]At-OTSA101		[92]
Advanced Refractory Solid tumors		[^225^Ac]Ac-FPI-1434	
Bladder Carcinoma		[^213^Bi]Bi-anti-EGFR mAb	[93]
Melanoma	[^225^Ac]Ac-crown-αMSH		[45]
Squamous cell carcinoma		^224^Ra (DaRT)	[94]

## Data Availability

No new data were created or analyzed in this study. Data sharing is not applicable to this article.

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
