# Peer review of "Targeted Alpha Therapy: Progress in Radionuclide Production, Radiochemistry, and Applications"

_pharmaceutics, 2020, doi:10.3390/pharmaceutics13010049_

Round 1

Reviewer 1 Report

The review paper written by Wuest et al., describes 'Targeted Alpha Therapy.' The content in this paper organizes well and it will provide useful information to scientists and medical doctors of related research fields. I would recommend 'accept' of this paper after some minor revisions. 

1) Resolution of figure 1 should be improved. 

2) If possible, add a figure showing key concept of targeted alpha therapy (production, radiolabeling and application). 

3) Please add molecular structure of some important chelators for labeling of radioactive metals.  

Reviewer 2 Report

In this paper, the authors analyze in detail the state of the art of targeted alpha therapy (TAT). Despite this is not the sole existing review of the literature on this topic, I appreciated the great effort made by the authors to encompass chemical, radiopharmaceutical, physical, preclinical, and clinical aspects of this fascinating prospect of nuclear medicine. The final result is a complete, well-written paper that will be of potential interest for educational purposes.

I have no methodological considerations. I just propose a few minor considerations to further improve the readability of this nice manuscript:

1) In its present form table 1 is not easily readable. Please modify the layout to make the title of each column more easily distinguishable from the remaining table content. Similarly, each kind of radioactive particle should be more clearly divided from others.

2) Due to its clinical content, perhaps paragraph 5.3 should be excluded from chapter 5 and renumbered as chapter 6.

3) Ra223 is currently the sole alpha emitters largely used in clinical practice. Perhaps the corresponding paragraph describing its clinical use might be a bit enlarged. As an example, the recent EMA restriction of use (https://www.ema.europa.eu/en/medicines/human/referrals/xofigo) should be discussed.

Reviewer 3 Report

please take into account some minor revisions of your manuscript.
